



# An integrated high-resolution bathymetric model for the Danube Delta system

Lauranne Alaerts[1,2], Jonathan Lambrechts[3], Ny Riana Randresihaja[1,2], Luc Vandenbulcke[1], Olivier Gourgue[4], Emmanuel Hanert[2,3], and Marilaure Grégoire[1]

[1]Department of Astrophysics, Geophysics and Oceanography (AGO), ULiège, Liège, Belgium
[2]Earth and Life Institute (ELI), UCLouvain, Louvain-la-Neuve, Belgium
[3]Institute of Mechanics, Materials and Civil Engineering (IMMC), UCLouvain, Louvain-la-Neuve, Belgium
[4]Operational Directorate Natural Environment (OD Nature), Royal Belgian Institute of Natural Sciences (RBINS), Brussels, Belgium

**Correspondence:** Lauranne Alaerts (lauranne.alaerts@uliege.be)

**Abstract.** Acting as a buffer between the Danube and the Black Sea, the Danube Delta plays an important role in regulating the hydro-biochemical flows of this land-sea continuum. Despite its importance, very few studies have focused on the impact of the Danube Delta on the different fluxes between the Danube and the Black Sea. One of the first step to characterize this land-sea continuum is to describe the bathymetry of the Delta. However, there is no complete, easily accessible bathymetric data on all three branches of the Delta to support hydrodynamic, biogeochemical or ecological studies. In this study, we aim to fill this gap by combining 4 different datasets, three in the river and one for the riverbanks, each varying in density and spatial distribution, to create a high-resolution bathymetry dataset. The bathymetric data was interpolated on a hybrid curvilinear-unstructured mesh with an anisotropic Inverse Distance Weighting (IDW) interpolation method. The resulting product offers resolutions ranging from 2 m in a connection zone to 100 m in one of the straight unidirectional channel. Cross validation of the dataset underlined the importance of the data source spatial pattern, with average Root Mean Square Error (RRMSE) of 0.55 %, 6.3 % and 27.6%, for river segments covered by the densest to the coarsest dataset. These error rates are comparable to those observed in bathymetry interpolation in rivers with similar source datasets. The bathymetry presented in this study is the first unique, high-resolution, comprehensive and easily accessible bathymetric model covering all three branches of the Danube Delta. It will serve as an input in a hydrodynamic model of the Danube Delta, with the aim of better understanding the role of the Delta in the land-sea continuum between the Danube and the Black Sea. The dataset is available at https://doi.org/10.5281/zenodo.14055741 (Alaerts et al., 2024).

## 1 Introduction

The Danube, Europe's second-largest river, flows through ten countries and drains an extensive catchment area of ∼800,000 km² before emptying into the Black Sea, where it is the primary source of water and nutrients. About 110 km before reaching the coast, the river divides into three main branches: the Chilia, Sulina and Sfantu Gheorghe branches (Fig. 1). Between those three branches, spanning ∼ 5000 km², lies the Danube Delta (Romanescu, 2013; Bănăduc et al., 2023; Driga, 2008). The

Danube Delta is a complex system of lakes, channels and flood plains. It acts as a crucial buffer zone for water, nutrients and sediments between the Danube river and the Black Sea (Tockner et al., 2009; Cristofor et al., 1993; Suciu et al., 2002). The delta is also a biodiversity hotspot and holds substantial importance for local inhabitants, providing essential resources such as

drinkable water, fishing, aquaculture, agricultural lands, transport and recreational activities (Bănăduc et al., 2016, 2023; Lazar et al., 2022).

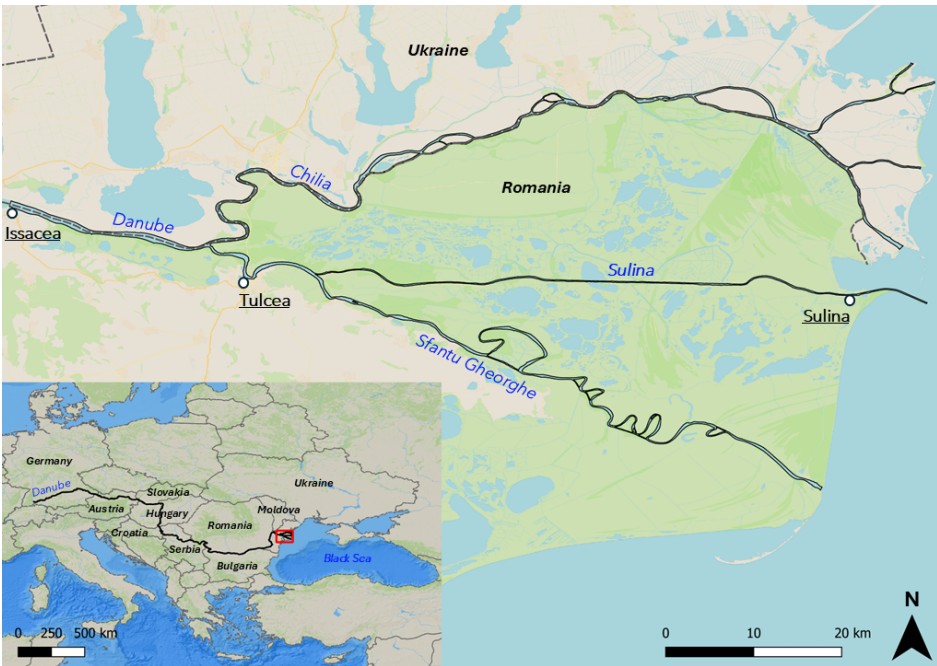

**Figure 1.** Map of the Danube Delta. The zoomed-out view (bottom left) displays the countries through which the Danube flows, with the river represented by a black line. The red rectangle outlines the area covered by the close-up view. In the close-up view of the delta, the black line marks the boundaries of the zone of interest, while the dashed line indicates the national borders. The basemaps are the Ocean Basemap (Esri, 2018) for the zoomed-out view, and for the close-up view the Voyager map tile by CartoDB, under CC BY 3.0. Data by OpenStreetMap © OpenStreetMap contributors 2019. Distributed under the Open Data Commons Open Database License (ODbL) v1.0.

Among the three branches, Chilia is the northernmost and serves as natural boundary between Ukraine and Romania. It is the youngest and least transformed of the three branches, and includes numerous meanders and islands. It divides into four smaller branches before reaching the sea, creating a secondary delta within the Danube Delta (Romanescu, 2013; Bănăduc et al., 2023).

The Sulina branch is located in the center of the delta. It is the shortest and most altered by anthropogenic activities of the three branches. The rectification of the branch, during a so-called 'cut-offs' program, took place at the end of the $19^{th}$ century and greatly impacted the path and discharge of the Sulina branch, now linking the cities of Tulcea and Sulina in an almost straight line. The branch is the main shipping route of the delta (Panin and Jipa, 2002; Driga, 2008; Duţu et al., 2018). Sfantu Gheorghe





is the southernmost branch. Like the Chilia branch, it meanders a lot, but an other 'cut-offs' program was carried-out in the
1980s to rectified all the meander bends, shortening the total length of the branch (Panin and Jipa, 2002).

Despite its ecological and socio-economic importance, comprehensive studies on the delta dynamics and ecological status
as a whole are scarce. In terms of bathymetry, some studies have focused on specific channels (Roşu et al., 2022; Jugaru Tiron
et al., 2009) and broader campaigns have covered entire branches (Duţu et al., 2018; DDNIRD, 2015). There is however no
publicly available bathymetry product covering the entirety of the three branches to support hydrodynamic, biogeochemical
or ecological studies on the Danube Delta as a whole. Bathymetry plays an important role in the description of aquatic envi-
ronment. Overly coarse resolution or poor-quality bathymetric data can result in substantial errors in predicting water column
heights, flood extents, velocities, and shear stress. In larger systems like deltas and estuaries, these inaccuracies can substan-
tially affect the distribution of water (Dey et al., 2022; Fuchs et al., 2022; Merwade et al., 2005).

Our goal in this study is to produce a complete bathymetry model of the three branches of the Danube Delta, from the city
of Issacea to the Black Sea (Fig. 1). To achieve that objective, we will use four different datasets that will be interpolated on
a hybrid curvilinear-unstructured mesh. The resulting product will subsequently be used in hydrodynamic models to better
represent the role of the delta within the Danube-Black Sea land-continuum.

## 2   Data and Methods

### 2.1   Data sources

We used data coming from four different sources. The first source is Copernicus' Digital Elevation Model (DEM) (European
Space Agency, 2021). We used it to determine the riverbanks' position and height. It has a resolution of 30 m, and the data dates
from 2021. The three other sources describe the bathymetry inside the river (Table 1). For the section of the Danube upstream
of the delta and the Chilia branch (represented in yellow in Fig. 2), the data come from measurements made by the Ukrainian
Scientific Center of Ecology of the Sea (UkrSCES) between 2014 and 2017. The data points were collected without adhering
to a specific sampling pattern, resulting in variable distances between points and inconsistent sampling density (Fig. 2.b.). The
averaged distance between two points is 160 m but can decrease down to$\sim$ 50 m. With 1925 data points, this dataset has an
averaged point density of $3.2 \times 10^{-5}$ points/m$^2$, which is rather low compared to datasets used in other studies (Merwade,
2009; Legleiter and Kyriakidis, 2008; Liang et al., 2022). The section upstream of the Sulina-Sfantu Gheorghe separation and
the Sulina branch (Fig 2.c.) are covered by data from the Galati Lower Danube River Administration (AFDJ). This data is
composed of measurements made on a regular grid of $\sim 1$ m resolution and dates from 2018. This dataset is composed of
$14.5 \times 10^6$ points, which gives a data point density of $0.4$ points/m$^2$, which is a very high density for a bathymetric survey
(Merwade, 2009; Legleiter and Kyriakidis, 2008; Liang et al., 2022). Data for the Sfantu Gheorghe branch comes from the
Danube Delta National Institute for Research and Development (DDNIRD) (DDNIRD, 2015). This data is made of transects
spaced 300 m apart on average (Fig 2.d.). The distance between two points within each transect is $\sim 3$ m. The measurement
campaign was carried out in 2015. There are $5.23 \times 10^4$ points in this dataset, and the point density is $1.7 \times 10^{-3}$ points/m$^2$.
This density is in the lower range of what is normally observed in bathymetry interpolation studies (Merwade, 2009; Legleiter





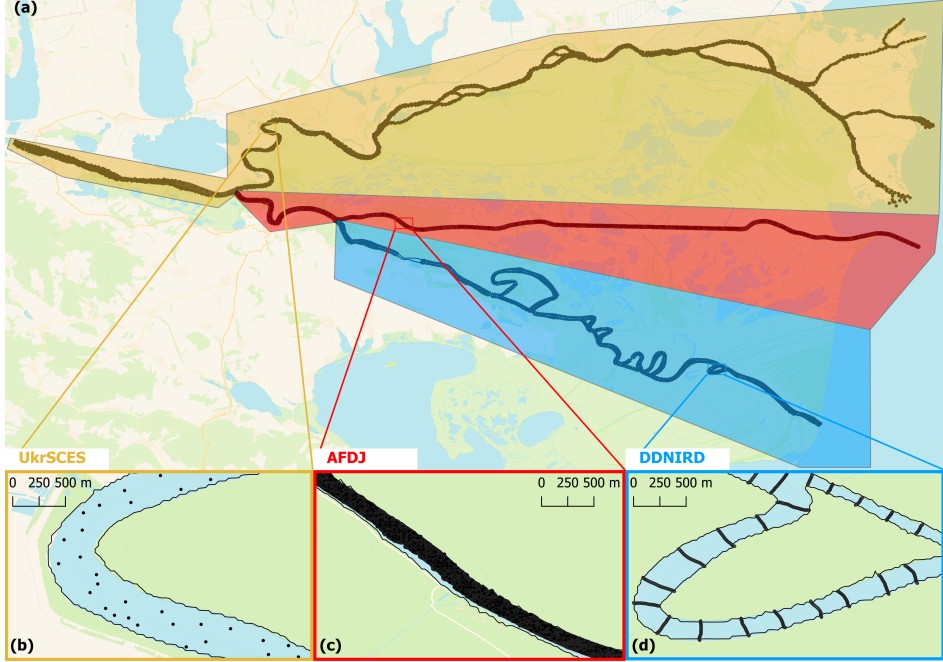

**Figure 2. (a)** Distribution of bathymetry sources within the delta. Each color represents a different source: UkrSCES in yellow, AFDJ in red, and DDNIRD in blue. Each black dot represents an individual bathymetry data point. **(b-d)** The bottom three panels provide close-up views at the same scale, displaying the bathymetry sampling points and highlighting the variations in sampling density among the three distinct bathymetry sources. The background image is the Voyager map tile by CartoDB, under CC BY 3.0. Data by OpenStreetMap © OpenStreetMap contributors 2019. Distributed under the Open Data Commons Open Database License (ODbL) v1.0.

and Kyriakidis, 2008; Liang et al., 2022). Due to data scarcity in the region, it was impossible to obtain bathymetric data of the same year for the entire domain.

## 2.2 Bathymetry interpolation

Given the diverse data sources, a certain degree of standardization was necessary. First, we had to transition most of the data from their local vertical datum to the WGS84 vertical datum. The UkrSCES data were referenced to the Odessa datum, which is 0.17 m below the WGS84 vertical datum. For AFDJ data, most of the Sulina channel was referenced to the Marea Neagra Sulina datum, 0.03 m above WGS84, while data upstream of Tulcea and the beginning of the Sulina channel used the Tulcea datum, 0.33 m above WGS84. DDNIRD data were referenced to the Marea Neagra 75 datum, 0.25 m above WGS84 (Anastasiu, 2014).

To merge the different bathymetry sources into a unified bathymetry product, a grid spanning the entire delta is required. In this study, we used a hybrid curvilinear-unstructured mesh (Fig. 3). The mesh consists of quadrilateral elements elongated along the flow in the unidirectional river segments (Figs. 3.b. and 3.d.), combined with unstructured triangular elements in the





**Table 1.** Characteristics of the different bathymetry data sources within the river.

| Source | Sampling strategy | No. of points | Density of points [points/m²] | Averaged distance between points [m] | Sampling years |
|--------|-------------------|---------------|-------------------------------|--------------------------------------|----------------|
| UkrSCES | Random points | 1925 | $3.2 \times 10^{-5}$ | 160 | 2014-2017 |
| AFDJ | Regular grid | $14.5 \times 10^{6}$ | 0.4 | $\sim 1$ | 2018 |
| DDNIRD | Transects | $5.23 \times 10^{4}$ | $1.7 \times 10^{-3}$ | $\sim 3$ (within a transect) | 2015 |
| | | | | 300 (between transects) | |

connection zones between segments (Fig. 3.c.). This configuration provides an accurate representation of both the river's course
and the bottom topography, while minimizing disk space requirements (Lai, 2010; Bomers et al., 2019). As the mesh elements
adapt to follow the shape of the river, the resolution is not constant. Perpendicular to the river, resolution varies between 2 and
12 m, with an average of 5 m. Along the river, element sizes range from 37 to 102 m, averaging 52 m. In the connection zones,
smaller elements are used, ranging from 2 to 9 m, with an average resolution of 5 m. Overall, mesh resolution varies from 2 m
(in a connection zone) and 102 m (on an edge along the riverbank). Further details on the mesh construction can be found in
Appendix A.

The data was then interpolated onto the generated mesh. River systems present unique challenges for interpolation due to
their inherent anisotropy: bathymetric variations tend to be more pronounced across the river, perpendicular to the flow, than
along its course. To address this, most present-day methods involve reprojection into a channel-centered coordinate system,
often referred as an $s, n$-coordinate system or curvilinear coordinate system. In this system, $s$ either represents the centerline
or the thalweg of the river, and $n$ its perpendicular. Once the bathymetric data is reprojected in the $s, n$-coordinate system, an
anisotropic interpolation is performed. In this study, we adopted this reprojection technique in combination with an anisotropic
Inverse Distance Weighting (IDW) interpolation method, as it is easy to implement and has shown good results in previous
studies (Merwade et al., 2006; Diaconu et al., 2019; Liang et al., 2022). Each unidirectional river segment was assigned its own
$s, n$-coordinate system, and interpolation was performed segment by segment. To ensure smooth transitions between adjacent
segments, the connections zones where included in the interpolation processes of each of their neighbouring segment. A
weighted mean was then calculated for each point of the connection zones, with weights inversely proportional to the distance
between the mesh point and the respective segment. Further details on the method can be found in Appendix B.





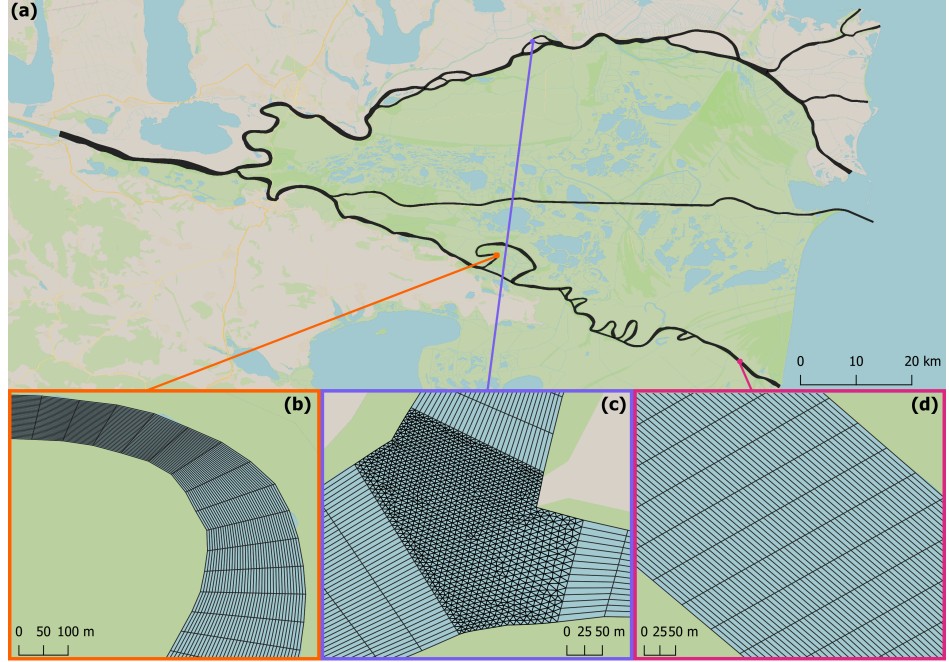

**Figure 3. (a)** Illustration of the unstructured mesh on the Danube, with zooms on **(b)** a bend in the river followed by a narrowing of the river,**(c)** an intersection between branches and **(d)** a straight portion of the river. The background image is the Voyager map tile by CartoDB, under CC BY 3.0. Data by OpenStreetMap © OpenStreetMap contributors 2019. Distributed under the Open Data Commons Open Database License (ODbL) v1.0.

## 2.3 Validation

### 2.3.1 Cross-validation

The lack of data in the Danube Delta means that there was no independent dataset to validate our bathymetry product. As a result, we chose to use a 'leave-one-out' cross-validation technique (Wu et al., 2019; Liang et al., 2022). In this method, one observation point is iteratively removed, and the interpolation is applied to estimate the bathymetry at that location. The estimated value is then compared to the actual observed data and this process is repeated for each point in the dataset. We chose this technique because it is best suited for datasets with low point density where every observation point is valuable.

It is however time-consuming since the process has to be repeated for each point of the dataset. This becomes particularly challenging with very large datasets, like the one from AFDJ, that contains $14.5 \times 10^6$ data points. To counter this problem, we chose to use a randomly selected subset of the points where the number of points is too high. As the validation is done segment by segment, we took a random sample of 1000 points for every segment that is covered by more than 1000 data points. In segments with less than 1000 points, all the points were used. Errors in the connection zones were calculated separately from

those in the segments. The error metrics we used in the validation are the Root Means Squared Error (RMSE) and Relative



Root Mean Squared Error (RRMSE):

$$RMSE = \sqrt{\frac{1}{n}\sum_{i=1}^{n}(z_{obs,i} - z_{predicted,i})^2}, \tag{1}$$

$$RRMSE = \frac{RMSE}{\frac{1}{n}\sum_{i=1}^{n} z_{obs},i} \times 100\%, \tag{2}$$

where $i = 1,...n$ are the $n$ points tested on the segment, $z_{obs,i}$ represents the observed value at the $i^{th}$ point, $z_{predicted,i}$ represents the interpolated bathymetry at the same point. The error in the connection was computed separately from the error in the segments, to be able to find the optimum combination method in the connection zone.

### 2.3.2 Comparison with global models

At present, there is no unique high-resolution bathymetry dataset easily available for the Danube Delta. In areas where such data is lacking, hydrodynamic models can use global bathymetry models as an alternative. To ensure that the bathymetry product developed in this study offers a clear improvement over existing resources, we compared it against two widely-used global bathymetry models: ETOPO 2022 (NOAA National Centers for Environmental Information, 2022) and GEBCO 2024 (GEBCO Compilation Group, 2024). Both models provide bathymetric data on a global scale, delivered on a grid with a resolution of 15 arc-seconds (~330 m).

## 3 Bathymetry Product

The final bathymetry product of this study covers the three main branches of the Danube Delta, including all the channels and meanders for which data were available, from Issacea to the Black Sea (Fig. 4). Detailed views of three areas, each covered by a different bathymetry data source, demonstrate the consistency between the interpolation and the observed data (Fig. 4.b-d). The dataset includes over $5.8 \times 10^5$ points. The bathymetry values range from -3.4 m (negative values indicate points above the reference level) on a dike, to 38.8 m in the river near Tulcea, with an average depth of $8.2 \pm 5.06$ m.

Our results align with general river morphology. The bathymetry displays anisotropic patterns, with more pronounced depth variations across the river than along its flow (Fig. 4 b., c. and d.). Greater depths are observed at the center of straight channels (Fig. 4 c.), and on the outer bends of the curves (Fig. 4 b. and d.). The meanders of the Sfantu Gheorghe branch tend to be shallower than the man-made straight channels created during the cut-off programs (Fig. 5). These patterns derive from hydrodynamic forces and sediment transport processes. Erosion tend to be more pronounced in areas with higher water velocities, such as outer bends and the center of straights channels. By contrast, sediment deposition is higher in slower-moving regions, like inner bends and meanders. In meandering channels, secondary helical flows, which transfer water from the outer bend near the surface downward toward the inner bend near the riverbed, further contribute to sediment accumulation in the inner bend (Bridge, 2003; Nelson et al., 2003).





Not all of the Sfantu Gheorghe branch meanders exhibit the same behaviours (Fig. 5). For example, while M1 and M3 meanders are indeed shallower than their respective artificial canals, the M2 meander displays the opposite pattern. Tiron Duţu et al. (2014) observed the same phenomenon, attributing it to the fact that the M2 meander has retained much of its activity, with its artificial canal being comparatively less active than those associated with M1 and M3. This unequal distribution of flow between the meander and the man-made channel is attributed to several geomorphological control factors, including the

channel length ratios, the diversion angle (i.e. angle between the main channel and the entrance of the diversion channel) and the bed level differences.

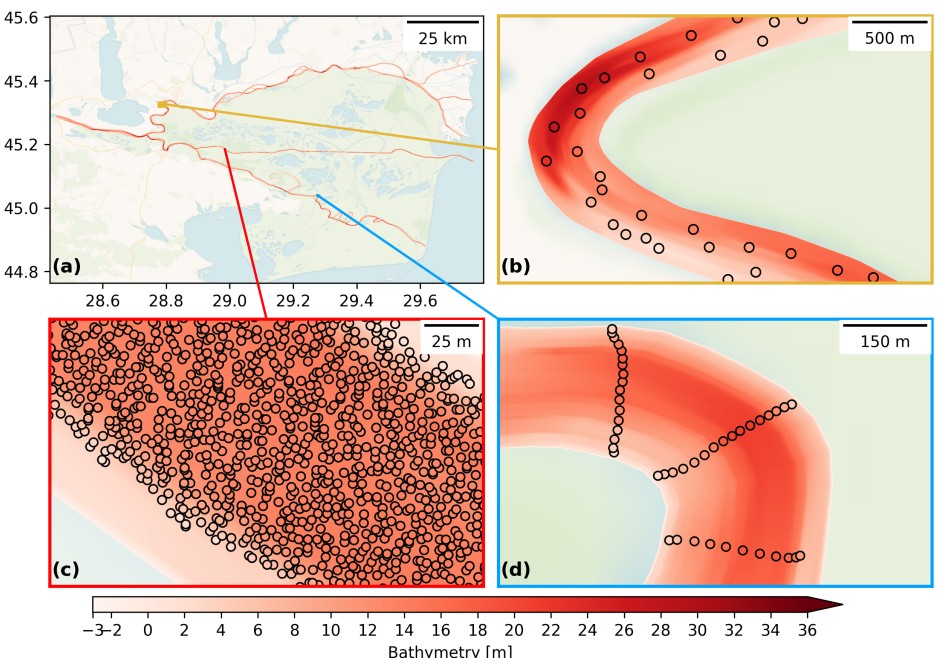

**Figure 4. (a)** Map of the interpolated bathymetry in the Danube Delta with close-up views on **(b)** the Chilia branch, with colored dots representing observation data, **(c)** the Sulina branch, with colored dots representing observation data, resampled to show 1/15 of the points for clarity, **(d)** the Sfantu Gheorghe branch, with colored dots representing observation data resampled to show 1/7 of the points for clarity. The background image is the Voyager map tile by CartoDB, under CC BY 3.0. Data by OpenStreetMap © OpenStreetMap contributors 2019. Distributed under the Open Data Commons Open Database License (ODbL) v1.0.

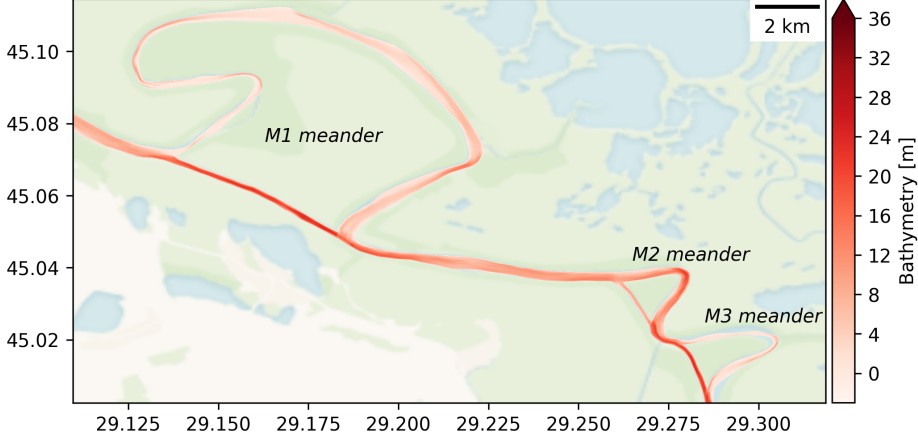

**Figure 5.** Map of the interpolated bathymetry in part of the Sfantu Gheorghe branch. M1 and M3 meanders present a shallower bathymetry than their respective artificial canal, while M2 meander is deeper than its man-made cut-off channel. The background image is the Voyager map tile by CartoDB, under CC BY 3.0. Data by OpenStreetMap © OpenStreetMap contributors 2019. Distributed under the Open Data Commons Open Database License (ODbL) v1.0.

## 4 Validation, applications and limitations

### 4.1 Validation

The RRMSE for each segment is strongly influenced by the primary bathymetric source (Fig. 6). Segments predominantly based on UkrSCES data show an average RRMSE of $27.6 \pm 13.4$ % (RMSE = 1.78 m). In contrast, segments primarily using AFDJ data exhibit a lower average RRMSE of $0.55 \pm 0.34$ % (RMSE = 0.07 m). In segments where the DDNIRD serves as the primary data source, the average RRMSE is $6.3 \pm 2.55$ % (RMSE = 0.5 m). In the connection zones, where a weighted mean was performed to ensure smooth transitions between segments, the average RRMSE is 1.90% (RMSE = 0.26 m).

Those results are comparable to those reported in studies using similar interpolation techniques. We did not find any studies that used datasets with a point resolution as low as that of the UkrSCES dataset. However, the RRMSE obtained for UkrSCES covered segments are similar to those of Merwade (2009) for two rivers with a random point distribution and a density approximately 3000 times higher than the UkrSCES dataset. For the DDNIRD dataset, our results are on par with or better than those found in the literature with transect and similar point densities (Merwade, 2009; Liang et al., 2022). We also found no studies using datasets with a point density as high as that of the AFDJ, but our RRMSE in AFDJ dominated segments is close to 0, indicating excellent alignment with the observed data. Additionally, our results conform to well-established river morphological patterns. They present a greater variation in depth across the river than along its flow, and deeper areas on the outer bends. The bathymetry in the Sfantu Gheorghe meanders reflects what has been observed by Tiron Duţu et al. (2014). As a result, we consider the bathymetric product to be of the highest possible quality with the available data sources.



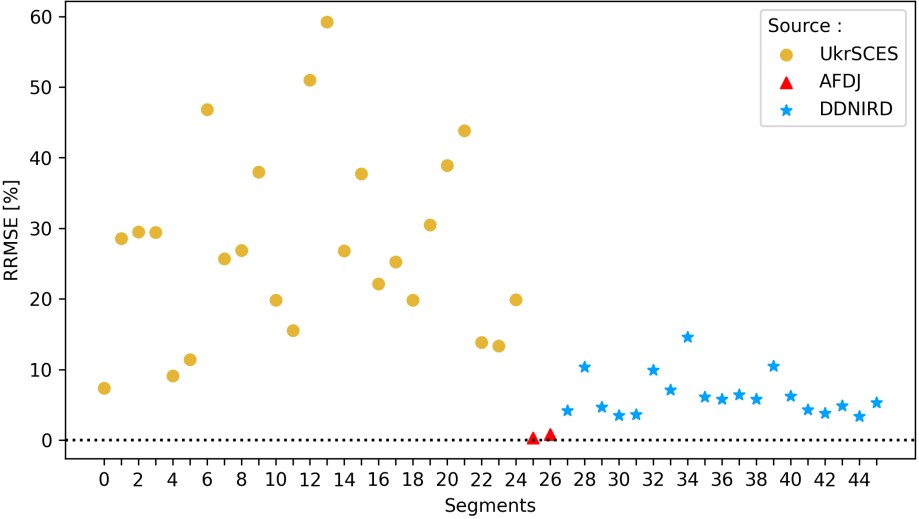

**Figure 6.** RRMSE for the bathymetry of each segment. The color and shape of the markers correspond to the main bathymetry data source covering the segment. The horizontal dotted line highlights the 0% RRMSE line.

## 4.2 Comparison with global bathymetry models

Our bathymetry product presents a significant improvement in representing the Danube Delta over global bathymetry models like ETOPO 2022 and GEBCO 2024 (Fig. 7). With a grid resolution of $\sim 330$ m, these global models struggle to represent the river bathymetry. In many cases, they either fail to capture the river's course or significantly underestimate its depth. Due to the coarse pixel size, which frequently exceeds the width of the river, they are unable to represent the depth variations within the river channel. While ETOPO 2022 and GEBCO 2024 are good bathymetry product for oceanographic applications, they lack

the necessary resolution to represent river processes. In contrast, our bathymetry product, which has a much higher resolution, is specifically tailored to capture these critical riverine dynamics, making it a more suitable tool for river-related studies and models.

## 4.3 Limitations

The most obvious limitation of this study comes from the data used. The first problem initiates from the temporality of

bathymetry samplings. Measurements were taken between 2014 and 2018. Readings for the AFDJ and DDNIRD datasets were each collected within a single year, in 2018 and 2015, respectively. This is not the case for the UkrSCES dataset, whose data spans three years (2014-2017). Rivers are very dynamic ecosystem, and the Danube is no exception, with both natural erosion/deposition of sediments and man-made dredging happening at different points along the river (Jugaru Tiron et al., 2009; Habersack et al., 2016; FAIRway Danube, 2021). It is therefore highly unlikely that the bathymetry remained the same during



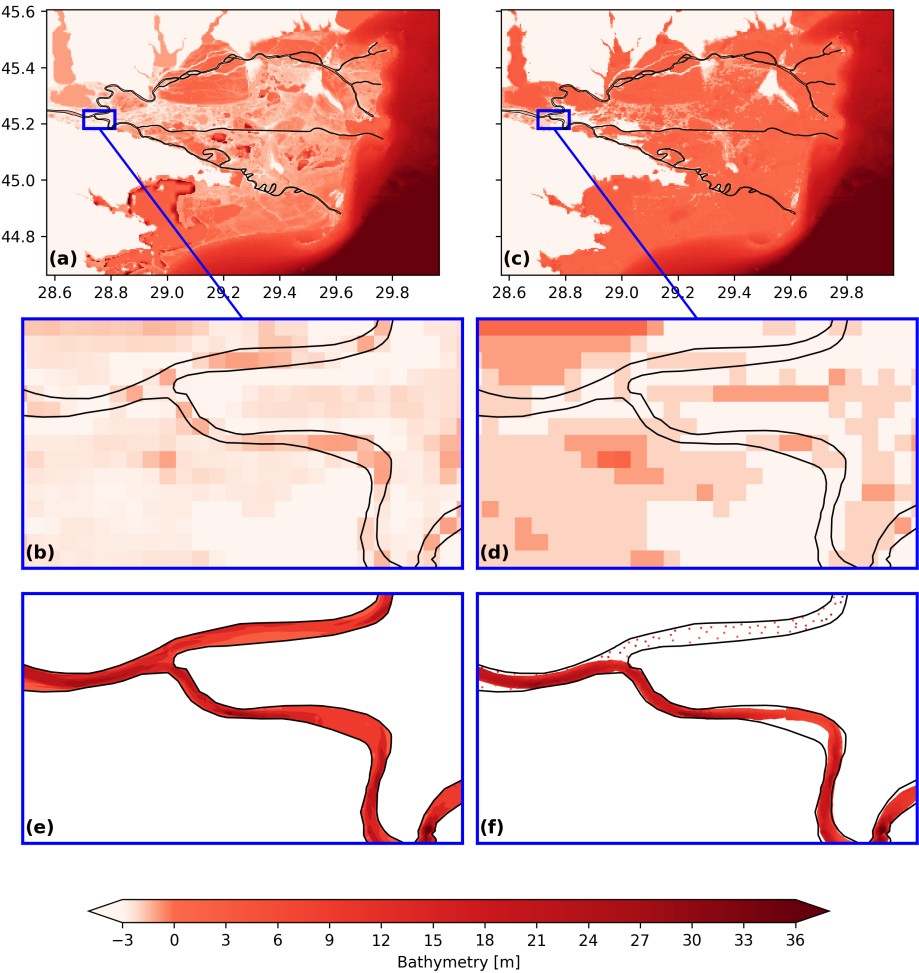

**Figure 7.** Comparison of ETOPO 2022 and GEBCO 2024 bathymetry of the delta with this study's results and observations. The Black line represent the Danube riverbanks. (a) Delta bathymetry according to ETOPO 2022 (NOAA National Centers for Environmental Information, 2022), with (b) a close-up view on ETOPO 2022 data in the Danube. (c) Delta bathymetry according to GEBCO 2024 (GEBCO Compilation Group, 2024), with (d) a close-up view on GEBCO 2024 data in the Danube. (e) Bathymetry product presented in this study. (f) Observations data points used in this study.

the entire sampling period. As such, it is only natural that we observed discrepancies at places where two different bathymetry sources meet.

The other limitation linked with the data is the low sampling density. More specifically, more data is needed for the river segments covered by the UkrSCES and DDNIRD datasets. For the UkrSCES dataset, the overall number of points is too low, and more data points should be taken, with special care to take points all across the width of the river. In the case of the

DDNIRD, the overall density is good but the spacing between transects is too high to ensure correct representation of the





continuity of the beds between measurements. To improve the quality of the results, the best solution would be to increase the density of points in the river, particularly for the UkrSCES dataset, and to a lesser extent for the DDNIRD dataset. For the former, any input of bathymetry point would be useful, as random-based bathymetry datasets can achieve performance comparable to transect-based data, if the density is high enough and points are correctly distributed across the width of the river (Merwade, 2009). For segments covered by the DDNIRD dataset, we suggest increasing transect frequency or adding longitudinal profiles parallel to the shores, as proposed by Diaconu et al. (2019).

The "by segment" interpolation used to allow reprojection in $s,n$-coordinate system is also a source of discontinuities between segments. The use of a weighted mean combination method in the connection zones reduced those discontinuities in the interpolation results and allow for more realistic transitions.

## 4.4 Applications

This work presents the first bathymetry dataset that comprehensively covers all three branches of the Danube Delta, marking a significant advancement in the characterization and understanding of the delta. This data set is an essential element in the development of a future hydrodynamic model of the Danube-Black Sea land-sea continuum. Such a model will improve our understanding of the complex coastal processes that occur in the Black Sea. Although the Danube Delta acts as a natural buffer between the river and the sea, most models studying the Danube's impact on the Black Sea do not investigate the river-delta-coastal zone continuum. These models typically represent the inputs of water and nutrients through boundary conditions by relying on monthly means or climatological averages. This approach oversimplifies the dynamic nature of the delta and its role in modulating freshwater and sediment fluxes (Grégoire and Friedrich, 2004; Beckers et al., 2002; Kara et al., 2008; Kubryakov et al., 2018; Lima et al., 2020). Such oversimplification can lead to inaccurate representations of hydrodynamic and biogeochemical conditions within the region where the freshwater from the Danube influences the Black Sea. This zone of influence can extend across large portions of the Black Sea shelf, especially from late spring to early fall. Poor resolution of the river input in the sea is one of the main source of error in coastal models, hampering their capabilities to predict the state and variability including the occurrence of extreme events like hypoxia (Ivanov et al., 2020; Breitburg et al., 2018; Bonamano et al., 2024; Rose et al., 2017). Therefore, having a high-resolution, easily accessible bathymetry dataset for the Danube Delta's branches is an important first step toward improving the accuracy of Black Sea coastal models and better understanding interactions within the Danube-Black Sea continuum.

Our bathymetry product could also be useful for flood risk assessment in the Danube Delta. The region is characterized by low elevations and minor altitude variations, with $\sim 93\%$ of its surface lying between 0 and 2 m above sea level. As a results, and despite the moderate rainfall in that area, the Danube Delta experience annual flooding (Niculescu et al., 2015). Coupled with a high-precision DEM, our bathymetry dataset could be used to represent the flooding processes within the delta. It could also support evaluations of infrastructure impacts, such as those of dikes and floodplain modifications, on flood extent and dynamics.

In addition, this bathymetry dataset can be used in ecosystem and habitat modeling. Designated a UNESCO World Heritage Site since 1990, the Danube Delta is Europe's largest nearly undisturbed wetland and is considered a major biodiversity hotspot





(Tockner et al., 2009; Simon and Andrei, 2023). Water depth, and by extension underwater topography, are key components of habitat suitability models. They influence many wildlife activities, including fish breeding, benthic communities distribution, reed bed development and bird nesting (Zhang et al., 2024; Zigler et al., 2008; Sultanov, 2019). High-resolution bathymetry can help pinpoint areas of ecological importance and guide conservation efforts.

## 5 Conclusions

This dataset is the first unique, high-resolution, comprehensive and easily accessible bathymetric model covering all three branches of the Danube Delta. We combined four different datasets of varying density and spatial patterns on a hybrid curvilinear-unstructured mesh using an anisotropic IDW interpolation method. The resulting product is made of $5.8 \times 10^5$ elements, with a resolution ranging from 2 to 100 m. Cross-validation confirmed that the error rates are comparable to those reported in similar interpolation studies, leading us to conclude that this product is as accurate as possible, given the avail-
able data. The dataset will be instrumental in enhancing hydrodynamic models aimed at improving our understanding of the Danube-Black Sea land-sea continuum. By offering better resolution and accuracy, this product will allow more precise simulations of river-coastal dynamics, providing essential insights for both scientific research and environmental management in the region.

## 6 Data availability

The dataset generated from this work is available at https://doi.org/10.5281/zenodo.14055741 (Alaerts et al., 2024)

## Appendix A: Mesh generation

To incorporate bathymetric data into hydrodynamic models, the river profile must be reconstructed on a mesh, ideally oriented along the flow direction (Merwade et al., 2005). In this study, we used an hybrid curvilinear-unstructured grid. It combines a curvilinear mesh made of quadrilateral elements elongated along the flow in unidirectional river segments, and an unstructured
triangular mesh at the connections between segments (Figures A1.d.). This hybrid curvilinear-unstructured grid has several advantages, the main one being that it allows an accurate representation of the river bottom with limited data storage (Lai, 2010; Bomers et al., 2019).

To create the mesh, the Danube is first divided into unidirectional segments and connection zones (i.e. zones where the river segments splits or merge) (Fig. A1.a.). To do so, the river is cut at a distance $L$ m from the points where the river segments intersect. We hence obtain 45 individual segments and 35 connection zones that will be meshed separately before being put
back together. The connection zones are meshed using triangular elements with a resolution of $l$ m (Fig. A1.b.). Each segment is then further subdivided into quadrilaterals, by cutting both riverbanks at $L$-meter intervals. To optimize this division of the





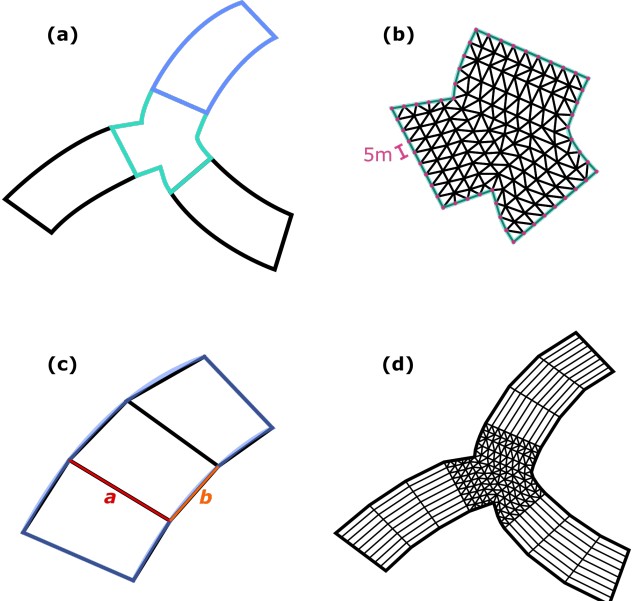

**Figure A1.** Creation of the mesh in the river. **(a)** The braided river is divided into unidirectional segments (black and blue) and connection zones (turquoise). **(b)** Connection zones are meshed with triangular elements at a resolution of approximately 5 m. **(c)** Segments are divided into quadrilaterals, where parameters $a$ and $b$ are optimized across each segment to ensure that the $b$ edges are as close to a chosen length as possible, and that $a$ and $b$ remain as perpendicular as possible (Eq. A1). **(d)** Segments are reassembled with connection zones. In each segment, each quadrilateral is further subdivided into smaller quadrilaterals aligned with the river's course to match the triangular elements of the connection zones.

segments, we calculate the following metric for each quadrilateral:

$$Q = 10(|\mathbf{b}| - L)^4 + (\mathbf{b} \cdot \hat{\mathbf{a}})^4, \tag{A1}$$

where $\mathbf{b}$ is a vector following the quadrilateral edge along the riverbank, $L$ is the target length of $\mathbf{b}$, and $\mathbf{a}$ is a vector following the edge of the same quadrilateral that serves as a cross section of the river (Fig. A1.c.). The quality metric $Q$ is computed for each combination of $\mathbf{a}$ and $\mathbf{b}$, and the sum of $Q$ is minimized for each segment. The aim of this optimization is to have quadrilaterals with $\mathbf{b}$ is as close as possible to the desired length $L$, and where the angle between $\mathbf{a}$ and $\mathbf{b}$ is as close as possible to 90°. In this way, we avoid having elements that are too small in curved river segments or near connections. The segments are

then assembled back together with the connection zones, and quadrilaterals in the segments are further subdivided into smaller ones, to ensure alignment with the 5-meter elements in the connection zones (Fig. A1.d.). The mesh was created using GMSH (Geuzaine and Remacle, 2009).

Our observation datasets have a resolution varying between a few meters and hundreds of meters. As a compromise, we chose to to fix $L = 50$ m and $l = 5$ m, obtaining a mesh made of 5 x 50 m elements in the segments and 5 m elements in

the connection zones. This resolution is coherent with the resolution of recent hydrodynamic models in rivers and deltas, that



generally have minimum element sizes varying between 5 to 50 m (Pham Van et al., 2016; Pelckmans et al., 2021; Bunya et al., 2010; Dresback et al., 2023; Bakhtyar et al., 2020). While this resolution may be coarser than the original bathymetric data in certain areas, particularly those covered by the AFDJ dataset, it provides a unified dataset with sufficient details to use in advanced hydrodynamic models while being more manageable than non-integrated higher-resolution datasets.

**Appendix B: Interpolation**

**B1    Interpolation process**

Given the diverse data sources, a certain degree of standardization was necessary. First, we had to transition most of the data from their local datum to the WGS84 vertical datum. The UkrSCES data were referenced to the Odessa datum, which is 0.17 m below the WGS84 vertical datum. For AFDJ data, most of the Sulina channel was referenced to the Marea Neagra Sulina datum, 0.03 m above WGS84, while data upstream of Tulcea and the beginning of the Sulina channel used the Tulcea datum, 0.33 m above WGS84. DDNIRD data were referenced to the Marea Neagra 75 datum, 0.25 m above WGS84 (Anastasiu, 2014).

Next, we interpolated the data onto the mesh. A first common step in river interpolation is to project the bathymetric data in a segment-oriented $s,n$-coordinate system (Merwade et al., 2005, 2006; Pelckmans et al., 2021; Legleiter and Kyriakidis, 2008). The reason behind this reprojection stems from the fact that conventional cartesian interpolation methods often fall short in providing satisfactory results in river settings. This inadequacy comes from the inherent anisotropy of riverbed bathymetry — the variation in river depth is notably more pronounced in the direction perpendicular to the flow compared to the flow direction itself. The initial projection of bathymetric data into an $s,n$-coordinate system allows us to address this anisotropy during the following interpolation. Here, $s$ represents the distance along the centerline of the river, while $n$ is the distance on the perpendicular to $s$.

Each segment defined during the mesh generation has its own coordinate system, based on the mesh of the segment. The projection procedure inside a segment is described below and illustrated in Fig. B1.

1. **Definition of the $s,n$-coordinate system:**

   The centerline of the river is computed, by determining the midpoint between every pair of nodes on each side of the river (Fig. B1.a.). This centerline subsequently serves as the $s$ axis. $s$-coordinates are allocated to each node along the riverbanks by calculating the length of the centerline from the beginning of the segment to the line connecting each pair of opposing nodes (Fig. B1.b.). All facing nodes within the same segment share identical $s$-coordinates. An $n$-coordinate is assigned to each node on the bank, by halving the distance between each pair of facing nodes (Fig. B1.c.). Nodes on the right bank have a negative $n$-coordinate, nodes on the left bank have a positive coordinate. As a result, each segment is subdivided into quadrilaterals, and each corner node within a quadrilateral receives coordinates within the $s,n$-system of the segment.





2. **Reprojection of the bathymetry points:**

For each bathymetry point, an assessment is made to determine the quadrilateral within which it is situated (Fig. B1.d.). Subsequently, its $s, n$-coordinate are calculated (Fig. B1.e.) with:

$$
\begin{aligned}
s &= \frac{1 - \delta_x}{2} s_{x_0} + \frac{1 + \delta_x}{2} s_{x_1}, \\
n &= \left( \frac{1 - \delta_x}{2} n_{x_0} + \frac{1 + \delta_x}{2} n_{x_1} \right) \frac{1 - \delta_y}{2} \\
&\quad + \left( \frac{1 - \delta_x}{2} n_{x_3} + \frac{1 + \delta_x}{2} n_{x_2} \right) \frac{1 + \delta_y}{2},
\end{aligned}
\tag{B1}
$$

where $\delta_x$ and $\delta_y$ are the coordinates of the bathymetry point in the quadrilateral, and $s_{x_i}$ and $n_{x_i}$ are the $s, n$-coordinates of the $i^{th}$ corner node of the quadrilateral.

To account for the river anisotropy during interpolation in the segments, one approach is to prioritize points with similar $n$-coordinate (i.e., directly upstream or downstream), over those with similar $s$-coordinates (i.e. on the same transect) (Merwade et al., 2006, 2008; Wu et al., 2019). To achieve this, we artificially increased the distance between bathymetry points in the direction perpendicular to the river by multiplying the $n$ coordinate of every point by an dimensionless anisotropy factor $an$. Subsequently, the depth at each node of the mesh was computed using Inverse Distance Weighting (IDW) interpolation:

$$
z^* = \sum_{i=1}^{np} w_i z_i,
\tag{B2}
$$

where $i = 1, ... np$ are the $np$ bathymetry points closest to the node, $z_i$ is the depth of the $i^{th}$ bathymetry point, and $w_i$ is the weight associated to this point:

$$
w_i = \frac{\frac{1}{d_i^p}}{\sum_{i=1}^{n} \frac{1}{d_i^p}},
\tag{B3}
$$

where $d_i$ is the distance between the node and the $i^{th}$ bathymetry point and $p$ is an exponent controlling the influence that the points have on the interpolation: the higher the power, the less distant points influence the depth value at the node. Similar methods, where the IDW interpolation is modified to take into account the anisotropy of the river, have given good results in previous studies, even faring better than other interpolation methods, such as kriging or spline interpolation (Merwade et al., 2006; Liang et al., 2022; Diaconu et al., 2019).

Another challenge in this study was handling interpolation in the connection zones, where multiple river segments converge or diverge. In theses areas, the deepest parts of the river do not follow a single, easily defined direction that could be approximated by a centerline $s$. Instead, the bathymetric features form complex patterns, often extending from the shapes of the adjacent segments and intersecting in "T" or "X" configurations. Few studies on river bathymetry interpolation focus on braided rivers with multiple river segments as their test cases. Goff and Nordfjord (2004) included the connections zones within the channels and took the maximum interpolated depth at points with multiple interpolation results. Hilton et al. (2019)



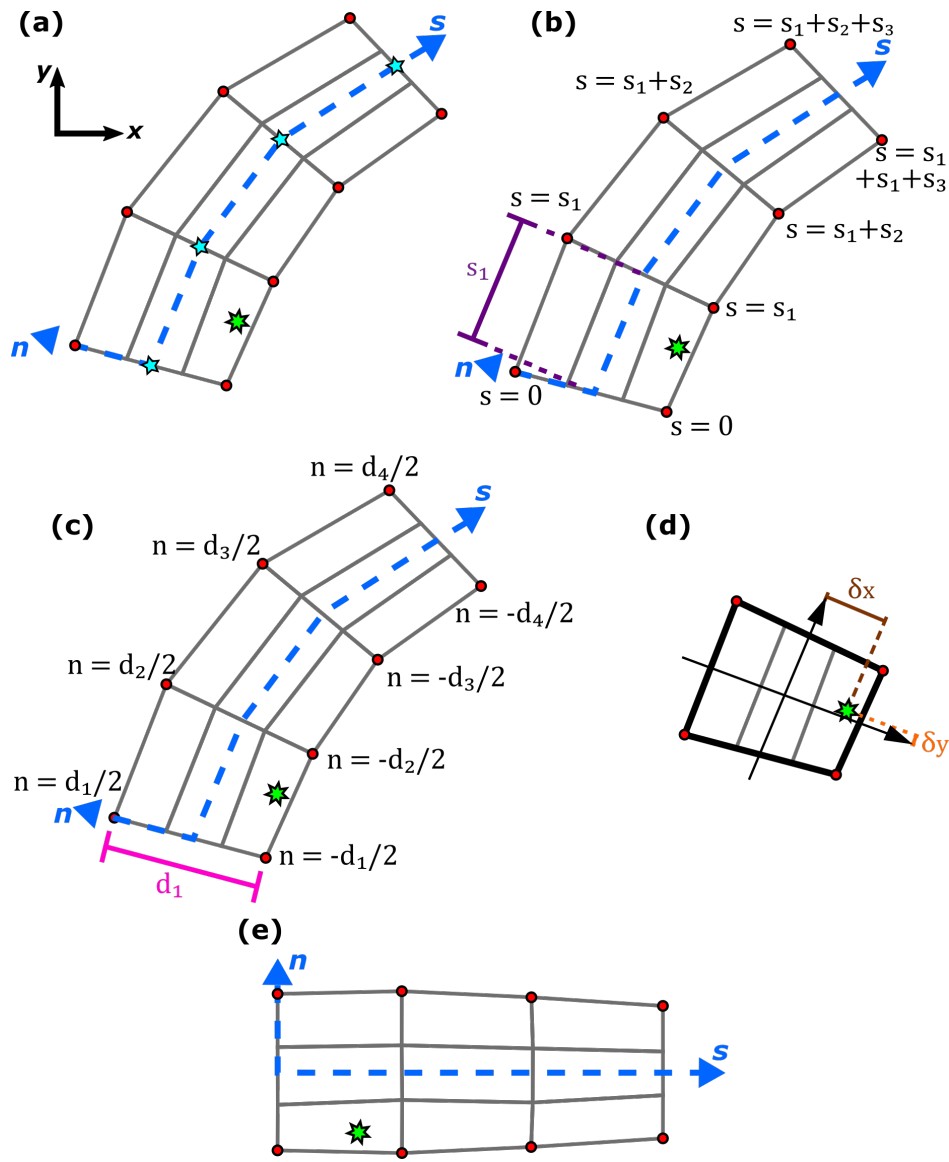

**Figure B1.** Reprojection of the bathymetry points in the $s, n$-coordinate system. The bathymetry point is represented by a green star. The nodes on the riverbanks are represented by red circle. The mesh is represented by gray lines. **(a)** We find the centerline (blue dotted line) of the river segment. The centerline passes in the middle (blue stars) of every pair of opposing nodes. **(b)** Every node on the riverbanks receives an $s$-coordinate. **(c)** Every node on the riverbanks receives an $n$-coordinate. **(d)** Coordinates of the bathymetry points in the quadrilateral in which it is located. **(e)** Bathymetry point in the $s, n$-coordinate system.

employed an $s, n$-coordinate system that encompassed the entire braided river network, with the $n$-coordinate spanning from 1 at the northernmost riverbank to -1 at the southernmost riverbank, thus avoiding the need to divide the network into different





channels. Similarly, Lai et al. (2021) did not cut the network in segments and opted to linearly interpolate the bathymetry
following the streamlines in the river network. Dey et al. (2022) did segmented the network, and computed the bathymetry
for points in the connection zones by using a 2- or 3-neighbors IDW, depending on whether or not they were on the tributary
side of the thalweg. While these methods produce satisfactory results, they also present limitations in terms of complexity or
compatibility with our domain. In this study, we chose to elongate the segment's mesh, to create a grid that extends into the
surrounding connection zones. This grid then served as a reference grid for reprojecting both the mesh points and bathymetric
data within the connection areas, following the procedure illustrated in Figure B1. Interpolation was then carried out as if the
mesh points of the connection zones included in this extended grid belonged to the segment. Consequently, a single mesh point
in a connection zone could be associated with multiple segment coordinate systems, leading to multiple interpolation results.
To determine the final bathymetry value for these points, we tested three different combination methods. The first method
followed Goff and Nordfjord (2004)'s approach, selecting the maximum value among the results. In the second method, we
calculated the mean of the values. In the third approach, we averaged the interpolation results, with weights assigned based
on the distance to the segment that produced each result. Those three methods are hereafter referred to as the Max, Mean and
Weighted mean methods, respectively.

## B2   Parametrization

A sensitivity analysis was carried out to identify the optimal values for the parameters $an$, $np$, and $p$ of the interpolation
method (Eqs. B2 and B3) in the river segments. For the anisotropy factor $an$, we tested integers between 1 and 20, followed
by multiples of 10 up to 100. The maximum value of 100 was chosen based on the DDNIRD data, where the distance within
a transect is approximately 3 m, while the distance between transects is about 300 m. The values tested for the number of
neighbors $np$ and the exponent $p$ are respectively integers between 2 and 8, and integers between 1 and 3. To select the best set
of parameters, we looked for the set of parameters that minimized the error obtained with the 'leave-one-out' cross-validation
technique. For segments with more than 1000 points, where a random sample of 1000 points was used for testing, the same
1000 points were tested for each parameter set.

   The optimal parameters for each segments are presented in Tables B1 and B2. Overall, we did not find any definitive rule to
define the optimum values of $an$, $np$ and $p$ parameters of the segments. However, certain trends can be observed regarding the
dimensionless anisotropic factor $an$. Segments covered by bathymetry primarily sourced from UkrSCES data tend to require
a higher $an$ value, with a median of 18. Segments relying on AFDJ or DDNIRD data have lower median $an$ of 1 and 2,
respectively. We observed no clear trend between the optimal values of the other parameters and the source of the observation
data. Segments mainly relying on AFDJ data appear to be insensitive to variations in parameter values.

   This initial parametrization worked well for sparse, randomly distributed bathymetric data and high-density bathymetry, such
as the data from the UkrSCES and the AFDJ. For the DDNIRD data, where points within transects are densely packed and
transects are widely spaced, this method led to interpolation errors on the higher-resolution grid. When optimizing parameters
in segments with DDNIRD data by sequentially removing points, the optimal $an$ value was often low. This is due to the fact
that points within the same transect are very close to each other and have similar bathymetry values, unlike the more distant





**Table B1.** Combination of parameters for the interpolation of the bathymetry that gives the lowest error for each segment covered by the UkrSCES dataset.

| Segment ID | $np$ | $p$ | $an$ | $RMSE$ [m] | $RRMSE$ [%] | Source |
|---|---|---|---|---|---|---|
| 0 | 3 | 2 | 14 | 1.21 | 7.35 | UkrSCES |
| 1 | 5 | 3 | 8 | 2.48 | 28.52 | UkrSCES |
| 2 | 7 | 1 | 18 | 2.10 | 29.50 | UkrSCES |
| 3 | 2 | 1 | 15 | 1.57 | 29.45 | UkrSCES |
| 4 | 2 | 3 | 90 | 0.60 | 9.09 | UkrSCES |
| 5 | 4 | 1 | 13 | 1.02 | 11.39 | UkrSCES |
| 6 | 2 | 3 | 30 | 1.54 | 46.81 | UkrSCES |
| 7 | 4 | 1 | 8 | 1.44 | 25.68 | UkrSCES |
| 8 | 3 | 3 | 6 | 2.69 | 26.89 | UkrSCES |
| 9 | 5 | 3 | 40 | 3.58 | 37.94 | UkrSCES |
| 10 | 6 | 2 | 14 | 1.54 | 19.84 | UkrSCES |
| 11 | 8 | 2 | 100 | 0.97 | 15.49 | UkrSCES |
| 12 | 3 | 3 | 100 | 2.09 | 51.00 | UkrSCES |
| 13 | 5 | 1 | 50 | 4.30 | 59.24 | UkrSCES |
| 14 | 5 | 2 | 100 | 1.57 | 26.81 | UkrSCES |
| 15 | 3 | 2 | 5 | 3.06 | 37.75 | UkrSCES |
| 16 | 4 | 2 | 20 | 1.25 | 22.10 | UkrSCES |
| 17 | 2 | 1 | 18 | 1.64 | 25.22 | UkrSCES |
| 18 | 4 | 2 | 15 | 1.50 | 19.82 | UkrSCES |
| 19 | 6 | 2 | 8 | 1.61 | 30.48 | UkrSCES |
| 20 | 4 | 1 | 1 | 1.29 | 38.89 | UkrSCES |
| 21 | 4 | 1 | 20 | 1.91 | 43.83 | UkrSCES |
| 22 | 4 | 1 | 30 | 1.04 | 13.82 | UkrSCES |
| 23 | 2 | 3 | 9 | 1.14 | 13.34 | UkrSCES |
| 24 | 3 | 2 | 18 | 1.44 | 19.89 | UkrSCES |

points in other transects. As a result, when the mesh resolution along the $s$-axis is finer than the distance between transects, the $np$ closest bathymetric points to the mesh nodes are often all from the same transect. This created a step-like interpolation, disrupting along-bed continuity.

To address this issue, we employed a modified two-step interpolation technique within the $s, n$-coordinate system for segments where DDNIRD data predominates. The first step is based on the idea pursued by several studies that the bathymetry changes linearly following lines of constant $n$-coordinates (Goff and Nordfjord, 2004; Caviedes-Voullième et al., 2014; Dysarz,



**Table B2.** Combination of parameters for the interpolation of the bathymetry that gives the lowest error for each segment covered by the AFDJ or the DDNIRD datasets.

| Segment ID | $np$ | $p$ | $an$ | $RMSE$ [m] | $RRMSE$ [%] | Source |
|---|---|---|---|---|---|---|
| 25 | 2 | 1 | 1 | 0.04 | 0.31 | AFDJ |
| 26 | 2 | 1 | 1 | 0.09 | 0.79 | AFDJ |
| 27 | 6 | 3 | 7 | 0.31 | 4.13 | DDNIRD |
| 28 | 2 | 1 | 14 | 0.32 | 12.66 | DDNIRD |
| 29 | 2 | 2 | 1 | 0.51 | 3.50 | DDNIRD |
| 30 | 3 | 2 | 1 | 0.42 | 5.09 | DDNIRD |
| 31 | 4 | 3 | 2 | 0.27 | 3.02 | DDNIRD |
| 32 | 3 | 2 | 1 | 0.35 | 6.72 | DDNIRD |
| 33 | 6 | 2 | 1 | 0.47 | 4.59 | DDNIRD |
| 34 | 2 | 1 | 4 | 0.14 | 9.79 | DDNIRD |
| 35 | 2 | 2 | 9 | 0.63 | 4.38 | DDNIRD |
| 36 | 4 | 2 | 2 | 0.40 | 4.15 | DDNIRD |
| 37 | 4 | 1 | 1 | 0.28 | 7.01 | DDNIRD |
| 38 | 6 | 2 | 40 | 0.64 | 4.47 | DDNIRD |
| 39 | 4 | 2 | 1 | 0.37 | 3.07 | DDNIRD |
| 40 | 4 | 2 | 1 | 0.19 | 5.88 | DDNIRD |
| 41 | 2 | 1 | 8 | 0.65 | 4.51 | DDNIRD |
| 42 | 2 | 1 | 13 | 0.27 | 2.76 | DDNIRD |
| 43 | 4 | 2 | 1 | 0.40 | 4.09 | DDNIRD |
| 44 | 2 | 1 | 3 | 0.29 | 2.95 | DDNIRD |
| 45 | 2 | 2 | 1 | 0.54 | 7.56 | DDNIRD |

2018). For each grid point of the grid used for reprojection, we computed the bathymetry by identifying the closest points from the upstream and downstream transects and then applying a simple Inverse Distance Weighting (IDW) interpolation using those two points in Eqs. B2 and B3, with $np = 2$ and $p = 1$. This process results in bathymetric data whose coordinates align with the mesh within the segment but not in the connection zones. To resolve this, a second interpolation is performed using the same method as with other bathymetry sources, but with the grid-interpolated bathymetry as the source. Since the grid and

the mesh coincide within the segments, the interpolated bathymetry in these areas remains largely unaffected by the second step, while the bathymetry on the mesh points in the connections continues smoothly from the segments. To estimate the error with this method, we used a "leave-one-out" cross-validation technique to compute the RRMSE on the observation points. It is important to note that this approach does not provide an error estimation for the mesh nodes between the transects.



**Table B3.** Error metrics in connection zones corresponding to the different combination methods described in Sec. B

| Combination method | RMSE [m] | RRMSE [%] | MAE [m] | MAD [m] |
|---|---|---|---|---|
| Max | 0.29 | 2.14 | 0.09 | 0.035 |
| Mean | 0.22 | 1.64 | 0.08 | 0.033 |
| Weighted mean | 0.26 | 1.90 | 0.08 | 0.032 |

In the connection zones, the three tested combination methods have similar errors (Table B3). They all gave low error metrics,
with at least 50% of the tested points with an absolute error below 3.5 cm. The Max method gives the poorest results. Although
the Mean method resulted in the lowest overall error, the Weighted Mean method provided smoother transitions between the
segments and has a lower MAD.

*Author contributions.*  LA, EH and MG designed the concept of the study. LA and LV acquired the data. LA and JL developed the methodol-
ogy. All authors provided input and suggestions throughout the project's progression. LA undertook the validation and wrote the initial draft
of the paper. All author contributed to the writing and editing of the manuscript.

*Competing interests.*  The authors declare that they have no conflict of interest.

*Acknowledgements.*  LA is a Ph.D. student supported by the Fond pour la formation à la Recherche dans l'Industrie et dans l'Agriculture
(FRIA/FNRS). NRR is a Ph.D. student supported by the Fonds de la Recherche Scientifique de Belgique (F.R.S.-FNRS). MG acknowledges
support from the EU H2020 BRIDGE-BS project under grant agreement no. 101000240 and from the Fond pour la formation à la Recherche
dans l'Industrie et dans l'Agriculture (FRIA/FNRS). The authors thank the project FULLCONTINUUM funded by the Fonds National de la
Recherche Scientifique and the BENTHOX program grant T.0102.21. Computational resources have been provided by the supercomputing
facilities of the Université catholique de Louvain (CISM/UCLouvain) and the Consortium des Équipements de Calcul Intensif en Fédération
Wallonie Bruxelles (CECI) funded by the Fonds de la Recherche Scientifique de Belgique (F.R.S.-FNRS) under convention 2.5020.11.



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
