# Peer review of "An integrated high-resolution bathymetric model for the Danube Delta system"

_Earth System Science Data, 2024_

## Author Comment (AC1)

**Reviewer 1:**

*(G) The collection of high-accuracy and robust bathymetric data is of paramount interest for any hydraulic and coastal study. Moreover, because of the complex morphology of river deltas, the bathymetry collection and interpolation are challenging. The paper presents well the analysis of different datasets and their interpolation in the river branches of the Danube Delta. The paper is well written and well structured, and results are clearly presented, with relevant figures and tables. I compliment the authors for their data collection and analysis work.*
*I recommend publication, subject to the authors addressing the comments made below.*

**Response:**

Thank you for your feedbacks! We sincerely appreciate your positive comments on our study. We have carefully considered and addressed the specific comments you provided and incorporated the necessary revisions.

(1) *I strongly recommend focusing abstract, section Application (4.4) and Conclusions on the topic of this dataset without digressing on future work on land-sea (modelling) studies (which can be the subject of another paper).*

**Response:**

As per your feedback, we modified the abstract, section "Application (4.4)" and conclusion to reduce the focus on the hydrodynamic model. We removed the sentence mentioning the model in the abstract, reduced the paragraph mentioning the hydrodynamic model in the application section (lines 204 to 212) and removed the sentence that talked specifically about the hydrodynamic model in the conclusion.

The first paragraph of the application section now reads: One of the possible applications for this dataset is its use in a hydro-biogeochemical model of the Danube-Black Sea continuum. The Danube Delta plays an important buffering role between the river and the sea, but most present-day models do not represent the delta (Beckers et al., 2002; Grégoire and Friedrich, 2004; Kara et al., 2008; Kubryakov et al., 2018; Lima et al., 2020).This oversimplification can lead to inaccuracies in the representation riverine inputs to the sea, which can in turn significantly impact the simulation of coastal processes (Bonamano et al., 2024; Breitburg et al., 2018; Ivanov et al., 2020; Rose et al., 2017). Therefore, having a high-resolution, easily accessible bathymetry dataset for the Danube Delta's branches is an important step toward improving of Black Sea coastal models and better understanding interactions within the Danube-Black Sea continuum. With that application in mind, future improvements to this dataset could include extending coverage to the shallow coastal waters in front of the delta.

(2) *Moreover, I suggest the author mention that such a dataset could be further improved including the bathymetry of the coastal area in front of the delta.*

**Response:**

Thank you for your suggestion! We decided to follow it by adding a sentence at the end of the paragraph about the Danube-Black Sea continuum model in the "Application (4.4)" section (see lines 211-212), as a coastal bathymetry would be an essential part for this application. This bathymetry product focuses on the Delta area where a consistent bathymetry dataset is

lacking. It could be extended with coastal bathymetric product (eg. EMODnet, GEBCO) using appropriate procedures.

It reads: With that application in mind, future improvements to this dataset could include extending coverage to the shallow coastal waters in front of the delta.

**References**

Beckers, J. M., Gregoire, M., Nihoul, J. C. J., Stanev, E., Staneva, J., and Lancelot, C.: Modelling the Danube-influenced North-western Continental Shelf of the Black Sea. I: Hydrodynamical Processes Simulated by 3-D and Box Models, Estuarine, Coastal and Shelf Science, 54, 453–472, https://doi.org/10.1006/ecss.2000.0658, 2002.

Bonamano, S., Federico, I., Causio, S., Piermattei, V., Piazzolla, D., Scanu, S., Madonia, A., Madonia, N., De Cillis, G., Jansen, E., Fersini, G., Coppini, G., and Marcelli, M.: River–coastal–ocean continuum modeling along the Lazio coast (Tyrrhenian Sea, Italy): Assessment of near river dynamics in the Tiber delta, Estuarine, Coastal and Shelf Science, 297, 108618, https://doi.org/10.1016/j.ecss.2024.108618, 2024.

Breitburg, D., Levin, L. A., Oschlies, A., Grégoire, M., Chavez, F. P., Conley, D. J., Garçon, V., Gilbert, D., Gutiérrez, D., Isensee, K., Jacinto, G. S., Limburg, K. E., Montes, I., Naqvi, S. W. A., Pitcher, G. C., Rabalais, N. N., Roman, M. R., Rose, K. A., Seibel, B. A., Telszewski, M., Yasuhara, M., and Zhang, J.: Declining oxygen in the global ocean and coastal waters, Science, 359, https://doi.org/10.1126/science.aam7240, 2018.

Grégoire, M. and Friedrich, J.: Nitrogen budget of the northwestern Black Sea shelf inferred from modeling studies and in situ benthic measurements, Marine Ecology Progress Series, 270, 15–39, https://doi.org/10.3354/meps270015, 2004.

Ivanov, E., Capet, A., Barth, A., Delhez, E. J. M., Soetaert, K., and Grégoire, M.: Hydrodynamic variability in the Southern Bight of the North Sea in response to typical atmospheric and tidal regimes. Benefit of using a high resolution model, Ocean Modelling, 154, 101682, https://doi.org/10.1016/j.ocemod.2020.101682, 2020.

Kara, A. B., Wallcraft, A. J., Hurlburt, H. E., and Stanev, E. V.: Air–sea fluxes and river discharges in the Black Sea with a focus on the Danube and Bosphorus, Journal of Marine Systems, 74, 74–95, https://doi.org/10.1016/j.jmarsys.2007.11.010, 2008.

Kubryakov, A. A., Stanichny, S. V., and Zatsepin, A. G.: Interannual variability of Danube waters propagation in summer period of 1992–2015 and its influence on the Black Sea ecosystem, Journal of Marine Systems, 179, 10–30, https://doi.org/10.1016/j.jmarsys.2017.11.001, 2018.

Lima, L., Aydogdu, A., Escudier, R., Masina, S., Cilibert, S. A., Azevedo, D., Peneva, E. L., Causio, S., Cipollone, A., Clementi, E., Creti, S., Stefanizzi, L., Lecci, R., Palermo, F., Coppini, G., Pinardi, N., and Palazov, A.: Black Sea Physical Reanalysis (CMEMS BS-Currents) (Version 1), https://doi.org/10.25423/CMCC/BLKSEA_MULTIYEAR_PHY_007_004, 2020.

Rose, K. A., Justic, D., Fennel, K., and Hetland, R. D.: Numerical Modeling of Hypoxia and Its Effects: Synthesis and Going Forward, in: Modeling Coastal Hypoxia: Numerical Simulations of Patterns, Controls and Effects of Dissolved Oxygen Dynamics, edited by: Justic, D., Rose, K. A., Hetland, R. D., and Fennel, K., Springer International Publishing, Cham, 401–421, https://doi.org/10.1007/978-3-319-54571-4_15, 2017.

---

## Author Comment (AC2)

**Reviewer 2:**

*(G) The study presents a well-structured and methodologically rigorous approach to high-resolution bathymetric modeling of the Danube Delta, effectively integrating multiple datasets and employing an adaptive interpolation framework to capture the complex morphology of the river system. The scientific methodology is sound, particularly in the use of a curvilinear, non-structured coordinate system combined with an anisotropic Inverse Distance Weighting (IDW) interpolation method, which enhances the spatial accuracy of the final dataset. Despite the inherent limitation of integrating multiple bathymetric databases, the authors adopt a sufficiently rigorous technical and scientific approach to ensure a consistent and methodologically robust expansion of the dataset over such an extensive and complex study area.*

**Response:**

Thank you for your feedbacks! We sincerely appreciate your thoughtful evaluation and recognition of our methodological approach. We are grateful for your constructive feedback and have carefully addressed the specific recommendations provided in our revised manuscript.

*(1) In my opinion, one of the primary limitations of the study lies in the relatively low resolution of the topographic data extracted from the Copernicus DEM (30m resolution). This may introduce uncertainties in the precise delineation of the shoreline and the transition between land and water. Given the importance of accurate topographic information for hydrodynamic and flood modeling applications, the use of a higher-resolution DEM could significantly enhance the reliability of the dataset. If such high-resolution topographic models are not available for this region, it would be beneficial for the authors to explicitly acknowledge this limitation in the text, clarifying whether finer-scale topographic data were considered but deemed unavailable or providing a justification for the choice of the Copernicus dataset.*

**Response:**

We acknowledge that a comment on the resolution of the Copernicus DEM was missing and thank you for your comment. In response, we have explicitly addressed this limitation in the paragraph about data resolution in the "4.3 Limitations" section (lines 173 to 185). This paragraph now clarifies that while a higher-resolution DEM could improve accuracy, to our knowledge, no such dataset currently exists for this region. Additionally, we moved this paragraph to the beginning of the "4.3 Limitations" section to enhance the logical flow of the discussion.

The revised first paragraph of the "4.3 Limitation" section now reads: The most obvious limitation of this study comes from the data used. The first problem is linked with the data resolution. In particular, additional bathymetric data would be beneficial for river segments covered by the UkrSCES and DDNIRD datasets, and a higher-resolution DEM could improve shoreline delineation. For the UkrSCES dataset, the overall number of points is too low, and more data points should be taken, with special care to take points all across the width of the river. In the case of the DDNIRD, the overall density is good but the spacing between transects is too high to ensure correct representation of the continuity of the bed between measurements. To improve the quality of the results, the best solution would be to increase the density of points in the river, particularly for the UkrSCES dataset, and to a lesser extent

for the DDNIRD dataset. For the former, any input of bathymetry point would be useful, as random-based bathymetry datasets can achieve performance comparable to transect-based data, if the density is high enough and points are correctly distributed across the width of the river (Merwade, 2009). For segments covered by the DDNIRD dataset, we suggest increasing transect frequency or adding longitudinal profiles parallel to the shores, as proposed by Diaconu et al. (2019). Concerning the topography data, the 30 m resolution of Copernicus' DEM may be insufficient to precisely define the riverbank positions. A higher-resolution DEM could improve accuracy, but to our knowledge, no such dataset is publicly available for this region.

(2) *Another key issue arises from the temporal mismatch between bathymetric and topographic data sources. River deltas are highly dynamic environments subject to morphological changes due to sediment deposition, erosion, and anthropogenic interventions, which may not be fully captured in the compiled dataset. The use of more temporally consistent datasets could improve the accuracy of hydrodynamic modeling. If no such datasets are available, the authors should explicitly state this limitation, specifying whether efforts were made to identify topographic data closer in time to the bathymetric surveys and explaining the rationale behind the chosen dataset.*

**Response:**

We acknowledge that the temporal mismatch between the topographic and bathymetric data introduces uncertainty, which was not explicitly addressed in the original manuscript. In response to your feedback; we have revised the paragraph discussing our data temporal disparity to include a comment on this issue (lines 186 to 197). The updated paragraph now explains that no Copernicus DEM product was available for the years corresponding to the bathymetry measurements, and that we selected the latest available DEM at the time of data collection. Additionally, to improve the logical flow of the discussion, we have repositioned this paragraph after the one addressing data resolution.

The revised paragraph now reads: The other limitation linked with the data initiates from the temporal disparity between datasets. The different measurements used in this study were taken between 2024 and 2021. The AFDJ and DDNIRD datasets were each collected within a single year, in 2018 and 2015, respectively. This is not the case for the UkrSCES dataset, whose data spans three years (2014-2017). The topography readings originate from the 2021 Copernicus DEM. Rivers are very dynamic ecosystem, and the Danube is no exception, with both natural erosion/deposition of sediments and man-made dredging happening at different points along the river(FAIRway Danube, 2021; Habersack et al., 2016; Jugaru Tiron et al., 2009). It is therefore highly unlikely that the bathymetry and position of the riverbanks remained the same during the entire sampling period. As such, it is only natural that we observed discrepancies at places where two different bathymetry sources meet. Despite this, the datasets used in this study were carefully selected to minimize temporal gaps while prioritizing the highest available spatial resolution and accessibility, ensuring the best possible representation of the riverbed for an easy-access bathymetry product. Regarding the Copernicus DEM, since no product was available for the years corresponding to the bathymetry measurements, we used the latest available dataset at the time of data collection, as it provided the best possible topographic coverage for the region.

(3) *In terms of readability, the paper is generally accessible, though some sections—particularly those describing interpolation techniques and coordinate transformations—could benefit from greater clarity and conciseness. While the technical language is appropriate, in some instances, overly complex phrasing may hinder comprehension for non-specialist readers. The use of English is strong overall, with only minor grammatical or stylistic issues that do not significantly affect readability.*

**Response:**

Thank you for your comments! As per your feedback, we modified the "Appendix B1 Interpolation process" section to make it more easily readable (lines 280 to 336). We also slightly modified the "Bathymetry interpolation" part of the method section (lines 84 to 96), to clarify the interpolation in connection zones and add references to the "Appendix B1 Interpolation process" section.

The modified paragraphs of the appendix section now read: Next, we interpolated the data onto the mesh. A first common step in river interpolation is to project the bathymetric data in a segment-oriented s, n-coordinate system(Legleiter and Kyriakidis, 2008; Merwade et al., 2005, 2006; Pelckmans et al., 2021). This projection improves interpolation results, as conventional cartesian interpolation methods often struggle to capture riverbed topography accurately because of the strong anisotropy of river systems. Depth variations are typcally much more pronounced across the river (perpendicular to the flow) than along its course. The initial projection of bathymetric data into an s, n-coordinate system allows us to accounts for this anisotropy in the following interpolation. In this study, s represents the distance along the centerline of the river, while n is the distance on the perpendicular to s.

Each river segment, as defined during mesh generation, has its own s, n-coordinate system. The projection process inside a segment is described below and illustrated in Fig. B1:

1. Definition of the s, n-coordinate system:
   The river centerline is computed as the midpoint between each pair of opposing bank nodes (Fig. B1.a.). This centerline serves as the s axis. The s-coordinate of each pair of opposing bank node is determined by measuring the centerline distance from the segment's starting point to the line connecting the two nodes (Fig. B1.b.). The n-coordinate is assigned to each node on the banks, by halving the distance between opposing nodes (Fig. B1.c.). Nodes on the right bank have a negative n-coordinate, nodes on the left bank have a positive coordinate. This results in a grid of quadrilateral elements, where each corner node has a new coordinate within the s, n-system of the segment.

2. Reprojection of the bathymetry points:
   Each bathymetry point is assigned to the quadrilateral it falls within (Fig. B1.d.). The s, n-coordinates of the bathymetry point are calculated (Fig. B1.e.) with
   $$s = \frac{1-\delta_x}{2} s_{x_0} + \frac{1+\delta_x}{2} s_{x_1},$$
   $$n = \left( \frac{1-\delta_x}{2} n_{x_0} + \frac{1+\delta_x}{2} n_{x_1} \right) \frac{1-\delta_y}{2} + \left( \frac{1-\delta_x}{2} n_{x_3} + \frac{1+\delta_x}{2} n_{x_2} \right) \frac{1+\delta_y}{2},$$
   where $\delta_x$ and $\delta_y$ are the point's local coordinates within the quadrilateral, and $s_{x_i}$ and $n_{x_i}$ are the s, n-coordinates of the $i^{th}$ corner node of the quadrilateral.

To account for the river anisotropy during interpolation in the segments, one approach is to give more weight to points with similar n-coordinate (i.e., directly upstream or downstream), than those with similar s-coordinates (i.e. on the same transect) (Merwade et al., 2006; Wu et al., 2019). To achieve this, we multiplied the n-coordinate of every points by a dimensionless anisotropy factor an to artificially increase the distance between bathymetry points in the

direction perpendicular to the river. Bathymetry values at mesh nodes were then computed using Inverse Distance Weighting (IDW) interpolation:

$$z^* = \sum_{i=1}^{np} w_i z_i,$$

where $z^*$ is the interpolated depth, $i = 1, \ldots np$ are the np bathymetry points closest to the node, $z_i$ is the depth of the $i^{th}$ bathymetry point, and wi is the weight associated to this point:

$$w_i = \frac{\frac{1}{d_i^p}}{\sum_{i=1}^{np} \frac{1}{d_i^p}},$$

where di is the distance between the node and the $i^{th}$ bathymetry point and p is an exponent controlling the influence that the points have on the interpolation. A higher p value reduces the effect of distant points on the interpolation. Similar methods, where the IDW interpolation is modified to take into account the anisotropy of the river, have given good results in previous studies, even outperforming other interpolation methods, such as kriging or spline interpolation (Diaconu et al., 2019; Liang et al., 2022; Merwade et al., 2006).

Another challenge in this study was handling interpolation in the connection zones, where multiple river segments converge or diverge. In these areas, the deepest parts of the river do not follow a single, easily defined direction that can be approximated by a centerline s. Instead, bathymetric features form complex patterns, often extending from adjacent segments and intersecting in "T" or "X" configurations. Few studies on river bathymetry interpolation focus on braided rivers with multiple river segments. Goff and Nordfjord (2004) included the connections zones within the river segments and took the maximum interpolated depth at points with multiple interpolation results. Hilton et al. (2019) employed an s, n-coordinate system that covered the entire braided river network, with the n-coordinate spanning from 1 at the northernmost riverbank to -1 at the southernmost riverbank, thus avoiding the need to divide the network into separate segments. Similarly, Lai et al. (2021) kept the whole river network and linearly interpolated the bathymetry along streamlines. In contrast, Dey et al. (2022) segmented the network, and interpolated points in connection zones using a 2- or 3-neighbor IDW approach, depending on whether the points were on the tributary side of the thalweg. While these methods produce satisfactory results, they also present limitations in terms of complexity or compatibility with our domain. In this study, we chose to elongate the segment's mesh, to create a grid that extends into the surrounding connection zones. This grid then served as a reference grid for projecting both the mesh points and bathymetric data within the connection areas, following the procedure illustrated in Figure B1. Interpolation was then performed as if the mesh points of the connection zones included in this extended grid belonged to the segment. As a result, each point in a connection zone could be assigned multiple bathymetry values from the interpolation in the different neighbouring segments. To determine the final bathymetry value for these points, we tested three different combination methods. The first method followed Goff and Nordfjord (2004)'s approach, selecting the maximum value among the results. In the second method, we calculated the mean of the values. In the third approach, we averaged the interpolation results with weights inversely proportional to the distance from the segment generating each result. Those three methods are hereafter referred to as the Max, Mean and Weighted mean methods, respectively

(4) *The figures and illustrations are well-designed and effectively support the text, providing valuable visual representations of the dataset, methodology, and results. However, some maps and graphs could benefit from higher contrast or improved labeling to enhance readability and interpretation.*

**Response:**

Thank you for your positive feedback on our figures! In response to your comment, we have adjusted the background of Figures 2, 3, 4, and 5 to enhance contrast and improve readability. Additionally, we have revised the captions of Figures 3 and 4 to provide clearer explanations for the reader.

The revised figures can be found hereafter:

[Figure]

*Figure 2. (a)* *Distribution of bathymetry sources within the delta. Each color represents a different source: UkrSCES in yellow, AFDJ in red, and DDNIRD in blue. Each black dot represents an individual bathymetry data point.* *(b-d)* *The bottom three panels provide close-up views at the same scale, displaying the bathymetry sampling points and highlighting the variations in sampling density among the three distinct bathymetry sources. The background image is the Voyager map tile by CartoDB, under CC BY 3.0. Data by OpenStreetMap © OpenStreetMap contributors 2019. Distributed under the Open Data Commons Open Database License (ODbL) v1.0.*

[Figure]

**Figure 3. (a)** *Illustration of the hybrid curvilinear-unstructured mesh on the Danube, with zooms on **(b)** a bend in the river followed by a narrowing of the river,**(c)** a connection zone between segments with its unstructured meshing and **(d)** a straight portion of the river. The background image is the Voyager map tile by CartoDB, under CC BY 3.0. Data by OpenStreetMap © OpenStreetMap contributors 2019. Distributed under the Open Data Commons Open Database License (ODbL) v1.0*

[Figure]

*Figure 4. (a) Map of the interpolated bathymetry in the Danube Delta, with close-up views (b-d) showing the interpolated bathymetry in the background and observations as colored dots. The modeled and observed bathymetry share the same colorbar in all the panels. (b) Close-up on the Chilia branch. (c) Close-up on the Sulina branch, where the observation data have been resampled to display 1/15 of the points for clarity. (d) Close-up on the Sfantu Gheorghe branch, where the observation data have been resampled to display 1/7 of the points for clarity. The background image is the Esri World Imagery basemap (ESRI, 2025)*

[Figure]

*Figure 5. Map of the interpolated bathymetry in a section of the Sfantu Gheorghe branch. M1 and M3 meanders present a shallower bathymetry than their respective artificial canal, while M2 meander is deeper than its man-made cut-off channel. The background image is the Esri World Imagery basemap (ESRI, 2025)*

(5) *Given the relevance and potential applications of this dataset, the authors could consider including, as a prospective future development, the dissemination of their dataset through WebGIS platforms. This could be explicitly discussed in the Discussion or Conclusions section as*

*a logical extension of their work. Implementing a WebGIS service would allow users to visualize, query, and analyze the dataset interactively, thereby significantly improving its accessibility and usability. Such an approach would expand the dataset's reach beyond the scientific community, making it a valuable resource for policymakers, environmental managers, and other stakeholders engaged in hydrodynamic modeling, flood risk assessment, and coastal zone management in the Danube Delta. By facilitating broader access and integration into decision-making processes, such an initiative would maximize the dataset's impact and further support interdisciplinary environmental assessments and planning efforts.*

**Response:**
We appreciate the suggestion to make the dataset available through a WebGIS platform. In response, we have added a paragraph in the "4.4 Applications" section (lines 225 to 232) highlighting the growing use of WebGIS tools for disseminating scientific datasets across various disciplines. We acknowledge that such a platform would enhance accessibility and usability, extending the dataset's reach beyond the scientific community to policymakers and environmental managers. While the implementation of a WebGIS system requires additional technical development and falls beyond the scope of this study, we recognize it as a valuable future step to maximize the impact of our dataset.

This new paragraph reads: While this dataset is currently only available through conventional repositories, future developments could focus on integrating it into an interactive WebGIS platform. WebGIS tools are getting increasingly used for disseminating scientific datasets in various fields (Dragićević, 2004; Foglini et al., 2025; Pasquaré Mariotto et al., 2021). They allow for an easy access and a larger dissemination of the information, providing broader accessibility beyond the scientific community that typically engages with data repositories. Web-based platform enable intuitive visualization, exploration, and interaction with the data, often incorporating tools for processing, analysis, and modeling. Although implementing such a system would require additional technical development and falls beyond the scope of this study, it could be a logical step toward enhancing the usability and impact of this dataset for a wider range of users, including policymakers, environmental managers, and researchers

**References**

Dey, S., Saksena, S., Winter, D., Merwade, V., and McMillan, S.: Incorporating Network Scale River Bathymetry to Improve Characterization of Fluvial Processes in Flood Modeling, Water Resources Research, 58, e2020WR029521, https://doi.org/10.1029/2020WR029521, 2022.

Diaconu, D. C., Bretcan, P., Peptenatu, D., Tanislav, D., and Mailat, E.: The importance of the number of points, transect location and interpolation techniques in the analysis of bathymetric measurements, Journal of Hydrology, 570, 774–785, https://doi.org/10.1016/j.jhydrol.2018.12.070, 2019.

Dragićević, S.: The potential of Web-based GIS, J Geograph Syst, 6, 79–81, https://doi.org/10.1007/s10109-004-0133-4, 2004.

FAIRway Danube: Fairway Rehabilitation and Maintenance Master Plan for the Danube and its navigable tributaries: NATIONAL ACTION PLANS UPDATE MAY 2021, Connecting Europe Facility of the European Union, 2021.

Foglini, F., Rovere, M., Tonielli, R., Castellan, G., Prampolini, M., Budillon, F., Cuffaro, M., Di Martino, G., Grande, V., Innangi, S., Loreto, M. F., Langone, L., Madricardo, F., Mercorella, A., Montagna, P., Palmiotto, C., Pellegrini, C., Petrizzo, A., Petracchini, L., Remia, A., Sacchi, M., Sanchez Galvez, D., Tassetti, A. N., and Trincardi, F.: A new multi-grid bathymetric dataset of the Gulf of Naples (Italy) from complementary multi-beam echo sounders, Earth System Science Data, 17, 181–203, https://doi.org/10.5194/essd-17-181-2025, 2025.

Goff, J. A. and Nordfjord, S.: Interpolation of Fluvial Morphology Using Channel-Oriented Coordinate Transformation: A Case Study from the New Jersey Shelf, Mathematical Geology, 36, 643–658, https://doi.org/10.1023/B:MATG.0000039539.84158.cd, 2004.

Habersack, H., Hein, T., Stanica, A., Liska, I., Mair, R., Jäger, E., Hauer, C., and Bradley, C.: Challenges of river basin management: Current status of, and prospects for, the River Danube from a river engineering perspective, Science of The Total Environment, 543, 828–845, https://doi.org/10.1016/j.scitotenv.2015.10.123, 2016.

Hilton, J. E., Grimaldi, S., Cohen, R. C. Z., Garg, N., Li, Y., Marvanek, S., Pauwels, V. R. N., and Walker, J. P.: River reconstruction using a conformal mapping method, Environmental Modelling & Software, 119, 197–213, https://doi.org/10.1016/j.envsoft.2019.06.006, 2019.

Jugaru Tiron, L., Le Coz, J., Provansal, M., Panin, N., Raccasi, G., Dramais, G., and Dussouillez, P.: Flow and sediment processes in a cutoff meander of the Danube Delta during episodic flooding, Geomorphology, 106, 186–197, https://doi.org/10.1016/j.geomorph.2008.10.016, 2009.

Lai, R., Wang, M., Zhang, X., Huang, L., Zhang, F., Yang, M., and Wang, M.: Streamline-Based Method for Reconstruction of Complex Braided River Bathymetry, Journal of Hydrologic Engineering, 26, 04021012, https://doi.org/10.1061/(ASCE)HE.1943-5584.0002080, 2021.

Legleiter, C. J. and Kyriakidis, P. C.: Spatial prediction of river channel topography by kriging, Earth Surf. Process. Landforms, 33, 841–867, https://doi.org/10.1002/esp.1579, 2008.

Liang, Y., Wang, B., Sheng, Y., and Liu, C.: Two-Step Simulation of Underwater Terrain in River Channel, Water, 14, 3041, https://doi.org/10.3390/w14193041, 2022.

Merwade, V. M.: Effect of spatial trends on interpolation of river bathymetry, Journal of Hydrology, 371, 169–181, https://doi.org/10.1016/j.jhydrol.2009.03.026, 2009.

Merwade, V. M., Maidment, D. R., and Hodges, B. R.: Geospatial Representation of River Channels, J. Hydrol. Eng., 10, 243–251, https://doi.org/10.1061/(ASCE)1084-0699(2005)10:3(243), 2005.

Merwade, V. M., Maidment, D. R., and Goff, J. A.: Anisotropic considerations while interpolating river channel bathymetry, Journal of Hydrology, 331, 731–741, https://doi.org/10.1016/j.jhydrol.2006.06.018, 2006.

Pasquaré Mariotto, F., Antoniou, V., Drymoni, K., Bonali, F. L., Nomikou, P., Fallati, L., Karatzaferis, O., and Vlasopoulos, O.: Virtual Geosite Communication through a WebGIS Platform: A Case Study from Santorini Island (Greece), Applied Sciences, 11, 5466, https://doi.org/10.3390/app11125466, 2021.

Pelckmans, I., Gourgue, O., Belliard, J.-P., Dominguez-Granda, L. E., Slobbe, C., and Temmerman, S.: Hydrodynamic modelling of the tide propagation in a tropical delta: overcoming the challenges of data scarcity, 2020 TELEMAC-MASCARET User Conference October 2021, Antwerp, 2021.

Wu, C.-Y., Mossa, J., Mao, L., and Almulla, M.: Comparison of different spatial interpolation methods for historical hydrographic data of the lowermost Mississippi River, Annals of GIS, 25, 133–151, https://doi.org/10.1080/19475683.2019.1588781, 2019.